# AI-Powered Synthesis of Structured Multimodal Breast Ultrasound Reports Integrating Radiologist Annotations and Deep Learning Analysis

**DOI:** 10.3390/bioengineering11090890

**Published:** 2024-09-01

**Authors:** Khadija Azhar, Byoung-Dai Lee, Shi Sub Byon, Kyu Ran Cho, Sung Eun Song

**Affiliations:** 1AI Laboratory, HealthHub Co., Ltd., Seoul 06524, Republic of Korea; khadija.azhar@healthhub.kr (K.A.); terry.byon@healthhub.kr (S.S.B.); 2Division of AI & Computer Engineering, Kyonggi University, Suwon 16227, Republic of Korea; blee@kgu.ac.kr; 3Department of Radiology, Korea University Anam Hospital, Korea University College of Medicine, Seoul 02841, Republic of Korea; krcho@korea.ac.kr

**Keywords:** breast ultrasound, deep learning, shear-wave elastography, report generation

## Abstract

Breast cancer is the most prevalent cancer among women worldwide. B-mode ultrasound (US) is essential for early detection, offering high sensitivity and specificity without radiation exposure. This study introduces a semi-automatic method to streamline breast US report generation, aiming to reduce the burden on radiologists. Our method synthesizes comprehensive breast US reports by combining the extracted information from radiologists’ annotations during routine screenings with the analysis results from deep learning algorithms on multimodal US images. Key modules in our method include image classification using visual features (ICVF), type classification via deep learning (TCDL), and automatic report structuring and compilation (ARSC). Experiments showed that the proposed method reduced the average report generation time to 3.8 min compared to manual processes, even when using relatively low-spec hardware. Generated reports perfectly matched ground truth reports for suspicious masses without a single failure on our evaluation datasets. Additionally, the deep-learning-based algorithm, utilizing DenseNet-121 as its core model, achieved an overall accuracy of 0.865, precision of 0.868, recall of 0.847, F1-score of 0.856, and area under the receiver operating characteristics of 0.92 in classifying tissue stiffness in breast US shear-wave elastography (SWE-mode) images. These improvements not only streamline the report generation process but also allow radiologists to dedicate more time and focus on patient care, ultimately enhancing clinical outcomes and patient satisfaction.

## 1. Introduction

Breast cancer is the most common cancer among women worldwide. It is characterized by an uncontrolled growth of cells within breast tissues. Based on recent findings by the Breast Cancer Research Foundation, approximately 2.3 million women were diagnosed with breast cancer worldwide in 2022, with 670,000 fatalities reported [1]. B-mode ultrasound (US) is a widely available, radiation-free, non-invasive, and effective imaging modality for early detection of breast cancer, with high sensitivity and specificity [2]. Recently, US elastography has become increasingly popular for aiding in the diagnosis and management of breast cancer [3,4,5]. This method allows for the quantification of tissue lesion stiffness and enhances traditional B-mode ultrasound in breast cancer screening.

In a typical breast US screening, the radiologist or sonographer thoroughly scans the entire breast area, including the underarm, to obtain detailed images of tissues. Any suspicious findings are saved for further evaluation. A radiologist then reviews these images, identifying signs of potential breast conditions such as cysts or tumors, and compiles a comprehensive report. This report includes characteristics of any detected abnormalities (such as size, shape, and echotexture). The radiologist may recommend additional diagnostic procedures such as a biopsy. Therefore, writing a precise, high-quality report requires not only extensive expertise in relevant fields, such as breast pathology, but also a substantial time commitment. For instance, experienced radiologists need 5–10 min on average to write a single US image report [6,7]. Incorporating US elastography results into B-mode US findings will further increase the workload of clinicians in report writing.

Recent advancement of deep learning technology, particularly in the field of natural language processing, has garnered significant interest in research on automatic generation of highly accurate medical reports. In early stages, models for generating medical reports primarily utilized recurrent neural networks (RNNs) such as long short-term memory (LSTM) and gated recurrent unit (GRU) [8,9,10]. However, with recent growth of large language models (LLMs), many studies are now proposing report generation models that demonstrate promising results by leveraging these advancements [11,12,13,14,15,16]. However, most of these studies have focused on standardized medical modalities such as magnetic resonance imaging (MRI), computed tomography (CT), and X-ray. Additionally, to be integrated into clinical workflows, improvements are required across multiple aspects including accuracy, reliability, and robustness of generated reports.

Therefore, the aim of this study was to provide a practical method that could be applied to the real clinical workflow to minimize the workload associated with breast US report writing. Rather than generating a breast US report fully automatically without human intervention, our approach focused on automating the generation of medical reports utilizing detailed annotations previously made on breast ultrasound images during routine screening procedures. Additionally, we enhanced the process by implementing deep learning algorithms to automatically classify tissue stiffness in breast US shear-wave elastography (SWE-mode) images. This method can reduce the time and effort typically required for manual breast US report writing while maintaining a high accuracy, effectively bridging the gap between fully manual and fully automated reporting (Figure 1).

Our main contributions can be summarized as follows:We introduced a semi-automatic method for generating breast US reports, leveraging existing annotations to blend automation with human oversight, thus reducing radiologists’ workload and maintaining report accuracy.We enhanced the semi-automatic report generation process by implementing deep learning algorithms specifically to classify tissue stiffness in breast US SWE-mode images.We conducted several experiments to validate the effectiveness of our semi-automatic report generation system, demonstrating its ability to accurately classify tissue stiffness and reduce report generation time, thereby proving its utility in clinical settings.

The remainer of this paper is organized as follows. Section 2 introduces recent research on ultrasound report generation using deep learning. Section 3 describes our proposed breast US report generation method in detail. Section 4 provides experimental results, detailing the effectiveness of the report generation. Finally, Section 5 discusses the implications of our findings and concludes the study.

## 2. Related Work

Recently, advances in artificial intelligence have led to the exploration of automated report generation in various ways, with models ranging in scale and complexity. Previous studies have focused on directly learning features from US scan images and associated raw medical reports, generating comprehensive reports using pre-trained large language models (LLMs) and/or custom-developed deep learning (DL) models. Zeng et al. [17] introduced a coarse-to-fine ultrasound image captioning ensemble model designed to help doctors automatically generate high-quality annotated ultrasound reports. The system detected organs using a coarse classification model, encodes the images with fine-grained classification based on organ labels, and then generates annotation text through a language generation model. Experiments showed that this approach achieves high accuracy in image recognition and produces high-quality annotation text, facilitating better understanding of ultrasound images for both patients and doctors.

The method proposed by Zeng et al. [18] used an object detection model to detect and encode focus areas in ultrasound images, addressing issues of low resolution and noise. Their system generated detailed annotation texts by inputting encoding vectors of focus areas into a language generation model. This end-to-end model captured more detailed information and improved language generation performance with reduced running time and parameters compared to full-size image captioning models.

In the work of Yang et al. [6], visual features of an input US image were first extracted and enhanced by a convolutional neural network and an attention mechanism. Resulting context vectors were then used by an LSTM to generate final reports. They demonstrated that their approach outperformed state-of-the-art image captioning methods with over 1 CIDEr score improvement. However, their approach has a drawback in that only the highest-quality image from US scan is used to extract key visual features.

As another recent example, Huh et al. [11] have proposed integrating multiple image analysis tools into breast reporting process for US scan images through a LangChain [19] using LLM. In their clinical evaluation with board-certified radiologists, they reported that the proposed system demonstrated satisfactory results in describing breast US images. Unlike our approach, where relevant US scan images were grouped together automatically, their rule-based image splitting process might not be able to handle cases where probe marks or main image positions differed significantly.

Qin et al. [20] proposed a novel report generation framework that combines unsupervised and supervised learning methods to align visual and textual features, inspired by the knowledge acquisition process of doctors. This framework utilized a similarity comparison mechanism to enhance the accuracy and comprehensiveness of generated reports by capturing global semantics without requiring additional disease labels. They validated the effectiveness of their approach using three distinct ultrasound datasets targeting the breast, thyroid, and liver, demonstrating its applicability across multiple organs.

Ge et al. [7] utilized an AI pipeline integrating personalized breast ultrasound screening reports, leveraging efficient preliminary report generation and allowing doctors to make adjustments based on AI-generated insights. The generated reports contained structured descriptions based on predefined categories, including the shape, internal echo pattern, and posterior acoustic features of the lesion.

In contrast to existing approaches, our study adopts a semi-automatic method for breast ultrasound report generation, integrating automation with human expertise. This approach offers a balanced solution that utilizes existing annotations to decrease radiologists’ workload while maintaining the integrity of report accuracy.

## 3. Materials and Methods

This retrospective study was approved by the Institutional Review Board (IRB) of Korea University Anam Hospital, where the research was conducted (IRB No. 2024AN0270). All methods adhered to ethical standards outlined in the Helsinki Declaration. The need for informed consent was waived by the IRB of our institution because data used in this retrospective study were already fully de-identified to protect patient confidentiality.

### 3.1. Study Population and Dataset

We prepared two datasets: one for developing deep-learning-based algorithms for automatic classification of breast US SWE-mode images, and another one for evaluating AI-assisted semi-automatic generation of breast US reports. For algorithm development, a total of 543 breast US SWE-mode Digital Imaging and Communication in Medicine (DICOM) images were collected from 543 non-overlapping patients using the picture archiving and communication system (PACS) at Korea University Anam Hospital. These images were scanned using GE’s LOGIQE10 (GE Healthcare, Illinois, USA) with a linear-type 18.5 MHz EUP-L75 probe between January 2023 and December 2023. Each SWE-mode image was manually annotated by a senior board-certified radiologist specializing in breast cancer, assigning one of three different classes (i.e., negative, positive, and equivocal classes) based on the tumor’s elasticity.

For model training and inference, DICOM images were converted into color images in PNG format while retaining their original resolution. Image resolution in the dataset was approximately 1292 × 970 pixels. Images were anonymized to protect privacy before use. The dataset was partitioned into training, validation, and testing data (59%, 323/543; 21%, 113/543; 20%, 107/543). The distribution of the dataset is shown in Table 1.

To evaluate automatically generated reports, we utilized patient data consisting of breast US DICOM images with corresponding data annotations as well as the final radiology report. This dataset contained scans from ten patients. These scans were randomly selected from the PACS at Korea University Anam Hospital. Breast US scanning time and equipment were the same as those used in the dataset for algorithm development. However, patients did not overlap. In addition, depending on the size, location, and number of lesions, not only the total number of DICOM files but also the scanned locations, angles, and directions for acquiring individual DICOM images varied for each patient. For instance, image data of one patient in the dataset included 58 DICOM images with different types of elastography images (e.g., Doppler elastography, shear-wave elastography, and strain elastography), whereas another patient’s data contained 35 DICOM images with only B-mode and SWE-mode images.

### 3.2. System Architecture

As shown in Figure 2, our proposed pipeline comprised three key modules, each contributing distinct functionalities: image classification using visual features (ICVF); type classification via deep learning (TCDL); and automatic report structuring and compilation (ARSC) modules. The following sections provides detailed descriptions of each module.

#### 3.2.1. Image Classification Using Visual Features (ICVF) Module

This module serves as the base as it analyzes US breast scans, systematically identifying their class based on several visual attributes. In this paper, we considered four types of US scans (i.e., B-mode, Doppler elastography, strain elastography, and shear-wave elastography). Examples of them are shown in Figure 3. Each type of scan has unique visual attributes that are specific to that type. For instance, in SWE-mode images, distinct characteristics emerged. Notably, the presence of a color bar with accompanying legends denoting units in kilopascals (kPa) and large colored contours were observed. SE-mode images also exhibited notable traits, with an absence of kPa unit indicators along the color legend bar, showcasing similarly large contours as observed in SWE-mode images. Conversely, Doppler images were distinguished by relatively smaller colored contour regions in comparison with SE-mode and SWE-mode images. Additionally, a notable feature was the presence of colored regions of interest (ROIs) that did not conform to a rectangular or square shape commonly found in SE-mode and SWE-mode images. These observations collectively aided the naked-eye identification and differentiation of visual attributes across different types of elastography images.

To classify US scan images based on their visual characteristics discussed above, an image processing algorithm is introduced, as shown in Algorithm 1. The algorithm first checks if the image is grayscale; if so, it categorizes the image as B-mode. For non-grayscale images, it assesses color pixel density and detects color contours to determine the mode. If the color pixel density is low and the contours are irregular, the image is classified as Doppler-mode. Conversely, if the text “kPa” is detected within the image around a colored legend bar, it is classified as SWE-mode. If neither condition is met, the algorithm defaults to SE-mode. This classification process utilizes a combination of color analysis, text detection, and contour examination to determine the ultrasound imaging mode accurately. Lastly, this module outputs an image annotated with its associated class, which is then passed as input to the type classification via deep learning (TCDL) module and automatic report structuring and compilation (ARSC) module for further parallel processing.
**Algorithm 1.** Image Classification Algorithm.**function** CLASSIFYIMAGE (image)   **if** image is grayscale **then**      **return** B-mode    **end if**    Calculate color pixel density   Detect Color Contours    Check for “kPa” text presence    **if** color pixel density is low AND Color Contours are irregular **then**      **return** Doppler-mode    **else**      **if** “kPa” text is present **then**       **return** SWE-mode      **else**       **return** SE-mode      **end if** **end if** **end function**

#### 3.2.2. Type Classification via Deep Learning (TCDL) Module

The TDCL module is composed of three sub-modules to process three types of elastography images. As a proof of concept, this study developed and validated a deep-learning-based algorithm for one of these image types (i.e., SWE-mode images). Essentially, the operational process within the TDCL module was the same regardless of image type. However, algorithms designed to process the three different image types might vary. By developing and validating the system for one image type in this study, we can verify the overall system’s effectiveness. If deemed effective, the proposed semi-automatic report generation system’s validity can be confirmed. The development of algorithms for the remaining two image types will be pursued in future research. Hereafter, the explanation is based on breast US SWE-mode images.

When the TDCL received an input from the ICVF module comprising both SWE-mode image and its associated scan type, it proceeded to extract ROIs from the input image, precisely isolating areas relevant to the identified scan type. Following this, ROIs were used as input to the deep learning model for further classification of classes for SWE-mode images. Detailed explanation for deep learning models will be presented later.

#### 3.2.3. Automatic Report Structuring and Compilation (ARSC) Module

The ARSC module is an integral part of automatic report generation. It begins with preprocessing, where it receives inputs from the TDCL module in the form of images, class, and associated scan types. As shown in Figure 4, the ARSC module consists of two main steps: Text Extraction and Report Compilation. In the Text Extraction, images from the ICVF module undergo pre-processing to isolate relevant text, located at the bottom third of DICOM scans. Hence, partial images containing only relevant textual parts were extracted from original US scans. The optical character recognition (OCR) algorithm then extracted textual information from these partial images. Thus, sorting and cleaning were needed to ensure accuracy. The location and size information were extracted using bounding box details, which might contain inaccuracies. A context-based correction approach was applied to correct these inaccuracies, such as replacing misclassified characters like “S” with “5”, “I” with “1”, “Z” with “2”, and “b” with “6” within size-related information. We developed a character confusion matrix to identify and correct common OCR misrecognitions, assigning context-based weights or ranks to each confusion pair based on their frequency. This matrix was developed during the testing and validation phase, comparing OCR outputs with ground truth data provided by radiologists. For instance, the correction of “S” to “5” is given higher weight due to its frequent occurrence, especially when “cm” follows, as “cm”, in our dataset, is consistently followed by a number. These corrections, weighted by their frequency, have significantly improved the OCR performance, reducing the character error rate (CER) from 25% to 0%. The cleaning process refined the extracted data, ensuring consistency and precision. Only decimal numbers were retained for size information. The second step, Report Compilation, involves grouping scans based on location data using inputs from TDCL Module. Afterwards, refined text and scan-type information were used to group scans based on location data. US scan images with identical location information were categorized into cohesive groups. Specifically, only side, clock face location, and depth details were considered for grouping purposes, disregarding other information. For example, images labeled “LEFT 2:00 5CM” and “LEFT 2:00 5CM LO” were grouped, ensuring different scans with the same location information were consolidated for streamlined analysis.

Then, the ARSC module formulates descriptive sentences for each group, encapsulating key finding. The template sentence was defined for the section of findings to ensure systematic and consistent reporting: “[size] mass in [location] from nipple, Doppler [class], Shearwave Elastography [class], Strain Elastography [class], Category [BI-RADS category]”. Contents in brackets are either automatically filled by the system or manually entered by a radiologist based on the review of breast US images. For instance, “size”, “location”, and “class” are automatically generated by analyzing a patient’s DICOM file. “BI-RADS category” represents categories 1 to 6 of the Breast Imaging Reporting and Data System (BI-RADS) [16]. It is determined by a radiologist analyzing images.

For each group of scan images, ARSC generates a descriptive report sentence using cleaned data of location and size information combined with a set of scan type information and corresponding class or type information predicted by the TCDL module. For instance, “0.41 × 1.2 cm mass in RIGHT 9 h, 6 CM from nipple Doppler (), Shearwave Elastography (positive), Strain Elastography (), Category ()”, where blank parentheses need to be filled by a radiologist. Formulated sentences are compiled into a text file, generating a comprehensive report for further analysis and interpretation. Each group’s sentence is saved for the radiologist’s convenience, streamlining tasks and ensuring accuracy. This approach enables radiologists to focus on delivering quality results without the burden of manual retrieval.

### 3.3. Deep Learning Models

#### 3.3.1. Optical Character Recognition (OCR) Model

OCR is a technology that enables the extraction of text from scanned documents or images, allowing systems to analyze and convert these documents into editable and searchable text automatically. This makes it easier to process, store, and retrieve information.

In our study, we used the OCR technology to extract annotated text in US scan images. We utilized a pre-trained model sourced from EasyOCR [21] based on a transformer architecture. EasyOCR comprises three key components: feature extraction, sequence labeling, and decoding (Figure 5). Feature extraction utilizes deep learning models such as ResNet [22] and VGG [23] to extract relevant features from input images, facilitating text recognition. Sequence labeling employs LSTM networks to interpret sequential context, vital for understanding text patterns. Decoding is accomplished using the connectionist temporal classification (CTC) algorithm, transcribing labeled sequences into recognized text. This cohesive framework allows EasyOCR to extract text from images effectively.

#### 3.3.2. Classification Model

For automatic classification of classes of breast SWE-mode images, we utilized the following four representative deep learning models for image classification task: ResNet [22], DenseNet [24], and EfficientNet [25]. ResNet utilizes a structured residual block with skip connections, accelerating learning and feature extraction while minimizing computational operations. The ResNet model employed in this study comprised 50 layers. The core aspect of ResNet is the residual block, which uses a skip connection that directly connects the input x to weight layers. Existing convolutional neural networks (CNNs) learn output values from input values. However, learning in ResNet is performed using a skip connection, as shown in Figure 6a, to minimize the value of *F*(*x*). This results in fast learning and avoids increasing the number of operations since the skip connection only adds a simple addition operation.

DenseNet structure allows for the stacking of information from the preceding layer and facilitates efficient transfer to the subsequent layer (see Figure 6b). This process enhances gradient flow, strengthens feature propagation, and promotes feature reuse without needing to relearn redundant features, thus reducing the number of parameters. In our implementation, we utilized a DenseNet model comprising 121 layers.

EfficientNet optimizes computational efficiency by scaling model depth, width, and resolution, balancing performance and computational cost. It incorporates techniques such as squeeze-and-excitation blocks [26] to better capture feature dependencies and reduce redundancy in feature maps. Additionally, EfficientNet uses a variant of the Inception module [27] called the MBConv block as shown in Figure 6c, which combines depth-wise separable convolutions with inverted residual connections to further optimize performance. In our implementation, we utilized EfficientNet-B6, which leveraged its efficient scaling strategy, incorporating squeeze-and-excitation blocks and MBConv blocks to enhance feature extraction and optimize model performance.

#### 3.3.3. Training Details

Our hardware configuration encompassed Ubuntu 22.04.4 LTS as the operating system. The CPU employed was an Intel(R) Core(TM) i5 8th Gen, coupled with an RTX 2060 GPU. Software-wise, we relied on Python 3.5, CUDA 10, and CuDNN 7.6.5, supplemented by TensorFlow 2.6.0 and Keras 2.1.0.

For model training, we employed TensorFlow 2.6.0 with DenseNet-121, ResNet-50, and EfficientNet-B6 architectures for classification tasks. Models were initialized with pre-trained weights and optimized using the Adam optimizer with an initial learning rate set at 0.001. A learning rate scheduler and early stopping callback were implemented to fine-tune the training process. Each model was trained for 50 epochs with a batch size of 8, gradually reducing the learning rate to ensure better convergence.

Input images were resized to 224 × 224 pixels while maintaining their aspect ratio. For DenseNet-121, we unfroze the last five layers for fine-tuning and optimized the learning rate to 0.0001. EfficientNet-B6 and ResNet-50 models were trained similarly with a learning rate of 0.001. Furthermore, our dataset exhibits class imbalance, as shown in Table 1, with the Negative class significantly underrepresented compared to the Equivocal and Positive classes. To address this imbalance, we assigned higher weights to the minority class during training. This ensures that misclassifications of the minority classes are penalized more heavily, encouraging the model to learn more effectively from these classes.

## 4. Results

### 4.1. Classification of Breast SWE-Mode Images

For performance analysis, several evaluation metrics were used. The corresponding equations are presented in Equation (1), where TP, FP, TN, and FN indicate true positive, false positive, true negative, and false negative, respectively. In addition, the area under the receiver operating characteristics (AUROC) curve was measured to evaluate the accuracy of the classification performance.
(1)$$\begin{array}{c} \mathrm{Accuracy}=\frac{\mathrm{TP}+\mathrm{TN}}{\mathrm{TP}+\mathrm{TN}+\mathrm{FP}+\mathrm{FN}}\\ \mathrm{Precision}=\frac{\mathrm{TP}}{\mathrm{TP}+\mathrm{FP}}\\ \mathrm{Recall}=\frac{\mathrm{TP}}{\mathrm{TP}+\mathrm{FN}}\\ F1-\mathrm{Score}=2\times \frac{\mathrm{Precision}\times \mathrm{Recall}}{\mathrm{Precision}+\mathrm{Recall}} \end{array}$$

To ensure robust evaluation, we applied a five-fold cross validation on the dataset. For each fold, we measured the corresponding performance metrics. The final result was the averaged result. Table 2 shows the overall performances of the three models. The DenseNet-121 exhibited the best performance across all metrics. For the assessment of model performance on a per-class basis, we analyzed the confusion matrix for individual deep learning models (Figure 7). For DenseNet-121, the classification accuracy for the positive class stood at an impressive 93%. Additionally, the negative class achieved an accuracy of 85%, while the equivocal class was accurately classified at 81%. Figure 8 shows AUROC curves of the three modes used in the experiments. In terms of AUROC, DenseNet-121 significantly outperformed other models. Specifically, it showed a 6.9% performance improvement over ResNet-50 and a 5.7% improvement over EfficientNet-B6.

### 4.2. Report Generation

As described in the previous section, US scan images of a patient sharing identical location and depth information were grouped together. Therefore, types of elastography included in each group might vary. The corresponding report for each group is then generated automatically if that group contains annotations made by a radiologist during the routine screening procedure. To evaluate the correctness of the proposed system, we measured the proportion of reports accurately generated by the proposed system among all reports written by a radiologist for suspicious masses. Specifically, we counted the number of reports where the location and depth of the suspicious mass as well as the type of elastography associated with the mass matched those in ground truth reports.

Results shown in Table 3 demonstrate the system’s performance in generating accurate reports for suspicious masses in ground truth (GT) reports across ten patients. In the analysis of a patient’s DICOM files, the total number of scan images is the total number of US images recorded. These images are then grouped based on their location and depth, resulting in the total number of groups. Within these groups, the system identifies suspicious masses reflected in the number of reports documented in the findings section of the GT report. Furthermore, the performance of the automated report generation system is evaluated based on the number of correctly generated reports, indicating matches with corresponding reports from GT reports.

The system demonstrated reliable accuracy in report generation for all 10 patients, achieving a perfect ratio of 1.0 by correctly generating all reports. For Patient 1, all seven reports were correctly generated. Patient 2 had all nine reports correctly generated. Patient 3 had all five reports correctly generated. Patients 4 through 10 also achieved perfect results, with the number of correctly generated reports matching the total number of reports for suspicious masses of each patient. This consistency is reflected in Table 3, which shows the proportion of correctly generated reports for each patient, highlighting the system’s reliability across different patient cases.

Building upon insights from the existing literature [6,7], which indicates the time-intensive nature of manual reporting by experienced radiologists, our study emphasizes the effectiveness of our semi-automatic approach in expediting the report generation process. By integrating deep learning algorithms and automated text extraction techniques, we significantly reduced report generation time. Specifically, our evaluations showed that the process for report generation for a patient averaged at 1.21 min, a notable reduction compared to traditional reporting norms, even when utilizing GPUs of relatively lower performance. This demonstrates the system’s robust efficiency regardless of high-end hardware dependency. Moreover, with better-performing hardware, the time required for report generation could be further reduced.

Figure 9 shows examples of automatically generated reports for individual groups of US scan images of a patient in the test dataset. The input used for these results consists of a total of 38 US scan images, with specific annotations for suspicious masses made at five out of the seven total locations. For each group, generated reports accurately captured the size and location of masses. Elastography findings were consistently identified.

## 5. Discussion and Conclusions

Several DL-based studies have utilized both B-mode and SWE-mode images to enhance breast cancer diagnostic performance [3]. However, to the best of our knowledge, this is the first study to analyze the feasibility of automatically classifying SWE-mode images based on their corresponding elasticity. We evaluated three representative DL models (i.e., ResNet-50, DenseNet-121, and EfficientNet-B6) for image classification using a self-constructed dataset. Among these models, DenseNet-121 showed the best performance (accuracy = 0.865; precision = 0.868; recall = 0.847; F1-Score = 0.856; AUROC = 0.923). While our dataset size is relatively small (543 images) and exhibits class imbalance, it is important to note that this imbalance reflects the real-world distribution of positive and negative cases in clinical settings. Despite these limitations, our findings suggest that advanced DL models for classifying SWE-mode images can potentially improve the accuracy and reliability of elasticity classification, thereby enhancing the overall diagnostic process for breast cancer. However, we acknowledge that a larger and more diverse dataset would be necessary to validate these results fully and ensure their generalizability across different clinical contexts.

Interpreting breast US images and writing an accurate diagnostic report are extraordinarily time-consuming and labor-intensive activities. The complexity of ultrasound scan images requires a high level of expertise and meticulous attention to detail to identify subtle pathological changes. Additionally, the necessity for precise documentation further compounds the workload, demanding significant cognitive effort and thoroughness from radiologists. In our study, instead of creating a breast US report entirely automatically without human involvement, as shown in previous studies, our approach aimed to automate the generation of medical reports by leveraging detailed annotations made on breast US images during routine screening procedures. This semi-automated method represents an intermediate step towards fully automated report generation, balancing human oversight with technological assistance. Overall, the proposed system demonstrated a high degree of accuracy in identifying the size and location of masses crucial for diagnostic purposes. In addition, types of US scan images are correctly identified, and elastography findings are consistently captured from input images. These efficiency gains underscore the potential for improving clinical workflows through timely and accurate medical reporting. We acknowledge that our current approach combines different methodologies, including EasyOCR and various CNNs. While this may not represent a single novel algorithm, it demonstrates the feasibility of integrating multiple cutting-edge technologies to address a complex clinical challenge. Future research could explore the incorporation of more advanced techniques such as large language models (LLMs) and attention mechanisms to enhance the system’s performance and novelty further. We recognize that for clinical applications, especially those assisting physicians, a highly stable and reliable system is crucial. Our current results, while promising, would require further validation and improvement to meet the stringent requirements for clinical use. Future work should focus on enhancing the system’s reliability, reproducibility, and stability.

This study has several limitations. First, we used only a single equipment model for breast US scan image acquisition. This limitation may affect the generalizability of our results to images obtained from different ultrasound machines, potentially leading to decreased performance when applied to diverse clinical settings. Second, we did not evaluate the proposed system or DL models using external datasets, which is crucial for ensuring the system’s reliability and generalizability. This lack of external validation limits our ability to claim, with confidence, the system’s effectiveness across different patient populations and clinical environments. Third, we did not perform a comprehensive analysis according to various factors such as demographic characteristics and lesion sizes. This omission may mask potential variations in system performance across different patient subgroups or lesion types, limiting our understanding of the system’s robustness. Fourth, our study was positioned as a proof of concept, exploring the integration of existing technologies rather than developing novel methodologies. Consequently, we employed well-established deep-learning-based algorithms that have proven effective in text extraction and image classification tasks on natural images, rather than algorithms specialized for breast ultrasound scan images. To address these limitations, future work will focus on collaborating with medical institutions to build a larger and more diverse dataset, allowing for extensive validation of our approach. In addition, we aim to enhance our system’s overall performance by developing specialized models tailored for breast ultrasound image analysis, incorporating cutting-edge technologies such as large language models and attention mechanisms in their development.

In conclusion, as an intermediate step toward the evolution of fully automated report generation, this study proposed a semi-automated approach that could blend human oversight with automation. This approach leverages existing annotations to generate breast reports, thereby reducing radiologists’ workload while maintaining high accuracy. Additionally, this study implements deep learning algorithms to classify tissue stiffness in breast US SWE-mode images, enhancing the report generation process. While our current results are promising, we recognize the need for further research and development to address the limitations identified in this study and to meet the rigorous standards required for clinical application.

## Figures and Tables

**Figure 1 bioengineering-11-00890-f001:**
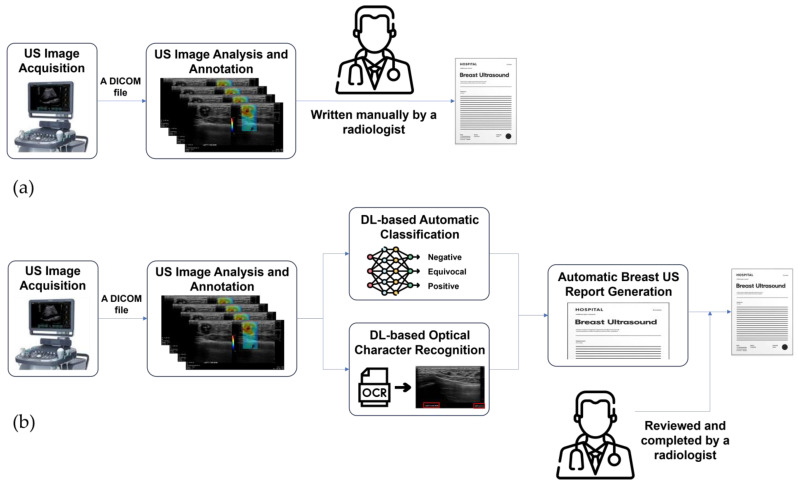
Schematic illustration of breast US report generation in the clinical US workflow (**a**) and the proposed AI-assisted semi-automatic generation of breast US reports (**b**).

**Figure 2 bioengineering-11-00890-f002:**
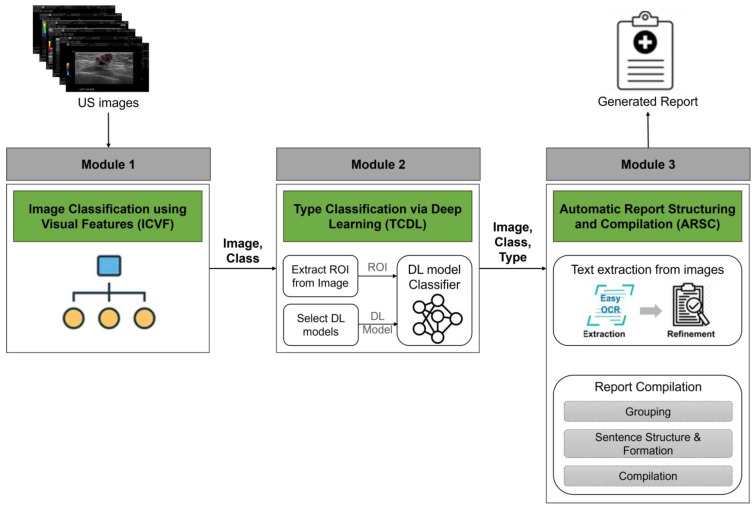
Our proposed pipeline consists of three modules: image classification using visual features (ICVF); type classification via deep learning (TCDL); and automatic report structuring and compilation (ARSC). ICVF analyzes ultrasound breast scans to identify classes based on visual attributes. TCDL focuses on classifying specific types within each class, while ARSC generates reports using text extracted from images.

**Figure 3 bioengineering-11-00890-f003:**
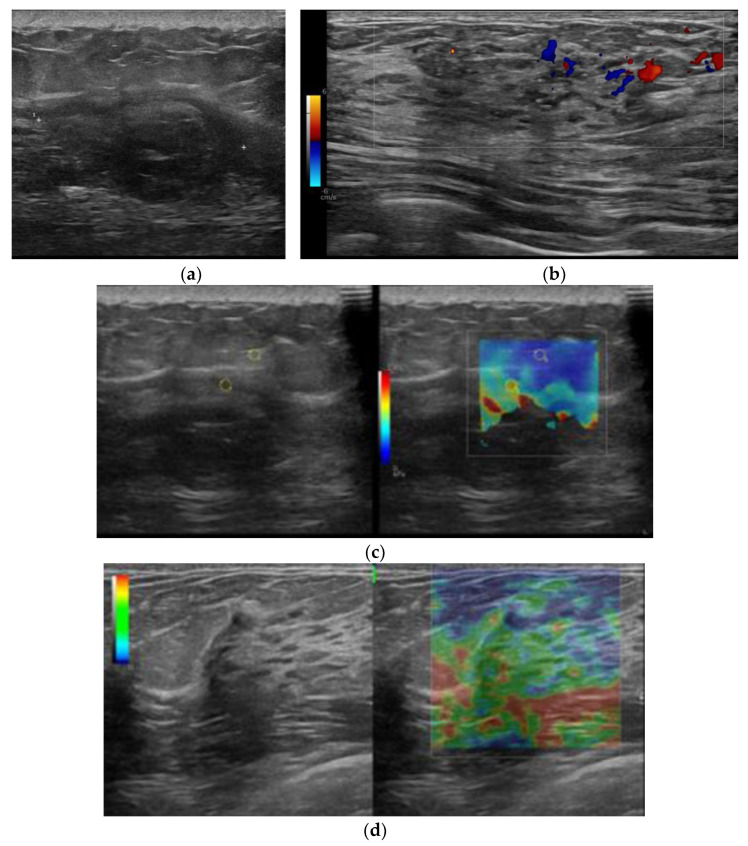
Four types of breast US scans images: (**a**) B-mode; (**b**) Doppler elastography; (**c**) shear-wave elastography; (**d**) strain elastography.

**Figure 4 bioengineering-11-00890-f004:**
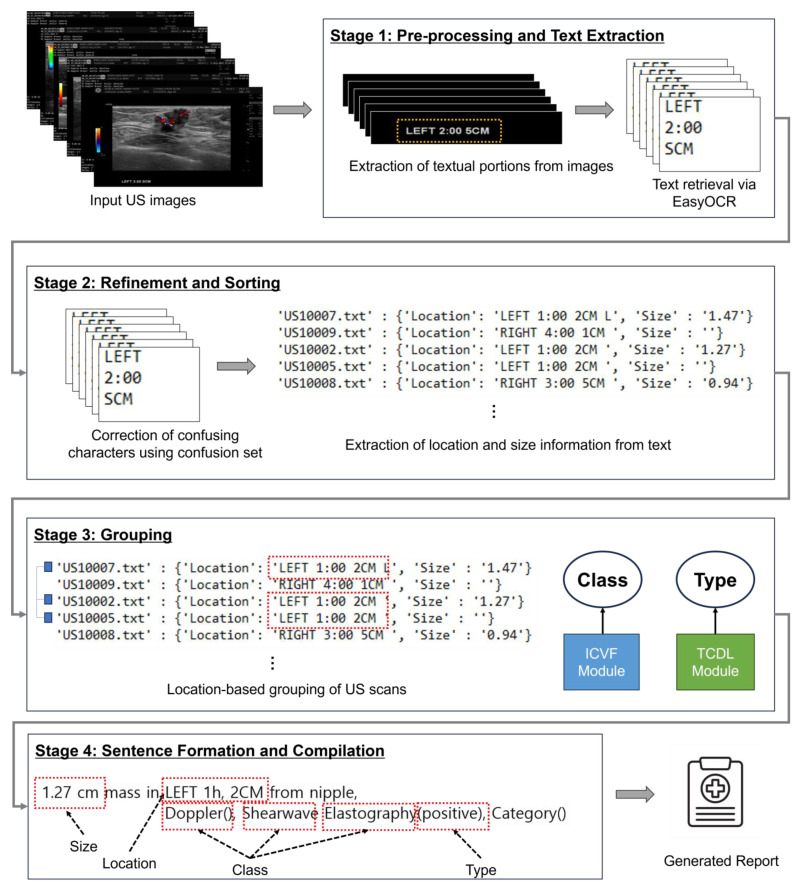
Comprehensive workflow of the ARSC module from US image input to automated structured clinical report synthesis. The red boxes represent the size, location, and class information extracted from annotations, along with the type information automatically predicted by the deep learning model.

**Figure 5 bioengineering-11-00890-f005:**
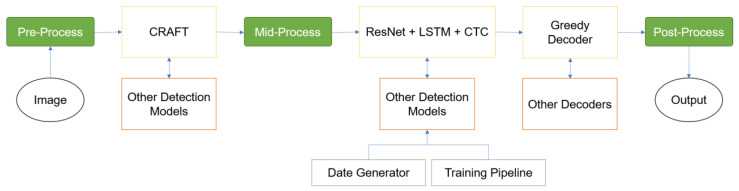
The pipeline of EasyOCR (excerpted from [21]).

**Figure 6 bioengineering-11-00890-f006:**
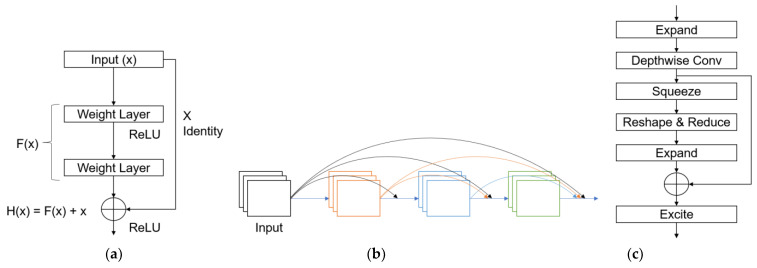
CNN core module: (**a**) residual block; (**b**) dense connectivity; and (**c**) EfficientNet module.

**Figure 7 bioengineering-11-00890-f007:**
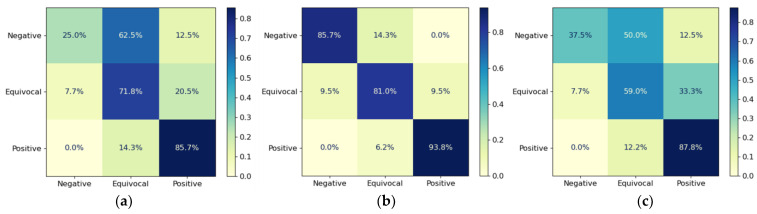
Confusion matrix (*X*-axis represents the predicted label and *Y*-axis denotes true labels): (**a**) ResNet-50; (**b**) DenseNet-121; and (**c**) EfficientNet-B6.

**Figure 8 bioengineering-11-00890-f008:**
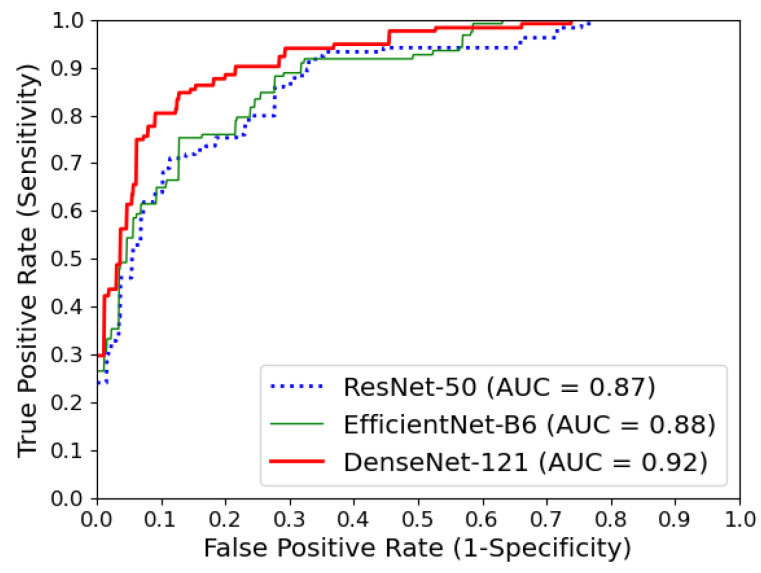
Area under the receiver operating characteristics (AUROC) curves for all models.

**Figure 9 bioengineering-11-00890-f009:**
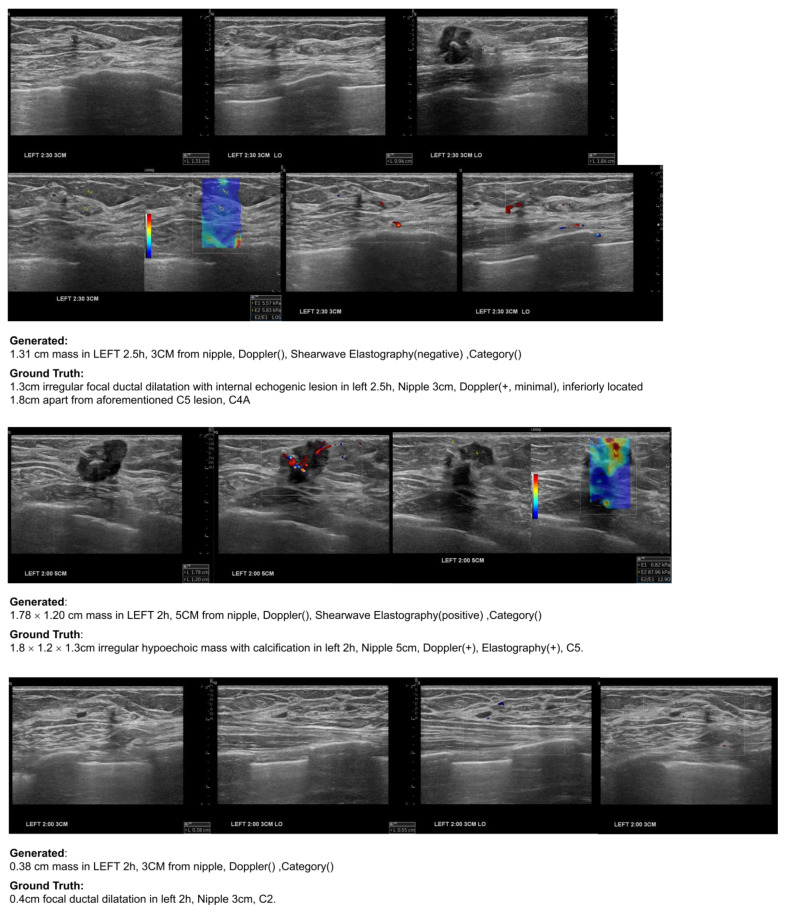
Example of report generation on breast US images of a patient in the test dataset. Each row corresponds to a group with manual annotations, the generated report sentence, and the ground truth report sentence.

**Table 1 bioengineering-11-00890-t001:** Distribution of the dataset for algorithm development.

Class	No. of Samples	Training Set	Validation Set	Test Set
Negative	81	48	17	16
Equivocal	198	118	41	39
Positive	264	157	55	52
Total	543	323 (59%)	113 (21%)	107 (20%)

**Table 2 bioengineering-11-00890-t002:** Performances of DL models.

Models	Accuracy	Precision	Recall	F1-Score
ResNet-50	0.411	0.570	0.320	0.345
DenseNet-121	0.865	0.868	0.847	0.856
EfficientNet-B6	0.711	0.706	0.655	0.672

**Table 3 bioengineering-11-00890-t003:** The proportion of correctly generated reports.

Patient ID	Total No. of Scan Images	Total No. of Groups	No. of Groups for Suspicious Masses	No. of Correctly Generated Reports	Ratio	Execution Time (min)
Patient 1	46	8	7	7	1.0	1.33
Patient 2	58	13	9	9	1.0	2.02
Patient 3	38	7	5	5	1.0	1.27
Patient 4	32	9	4	4	1.0	1.00
Patient 5	7	2	1	1	1.0	0.30
Patient 6	23	5	4	4	1.0	0.49
Patient 7	56	11	9	9	1.0	1.88
Patient 8	19	2	2	2	1.0	0.48
Patient 9	52	12	11	11	1.0	1.78
Patient 10	51	9	9	9	1.0	1.59

## Data Availability

According to the data policy of Korea University Anam Hospital where this research was conducted, permission from the Korea University Anam Hospital is required to export or disclose data. Therefore, administrative procedures of Korea University Anam Hospital must be followed in order for the researchers to provide data to external researchers or institutions. This approval process is exempt for research conducted by internal researchers. Thus, it was not needed for the current research. Consequently, we cannot provide data at this time. However, data are available from the corresponding author after completing the necessary procedures.

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
