# Peer review of "AI-Powered Synthesis of Structured Multimodal Breast Ultrasound Reports Integrating Radiologist Annotations and Deep Learning Analysis"

_bioengineering, 2024, doi:10.3390/bioengineering11090890_

Round 1

Reviewer 1 Report (Previous Reviewer 1)

Comments and Suggestions for Authors

My three major concerns were 1) very small dataset (only 543 images), 2) serious class imbalance (only 81/543 (around 15%) samples are negative), and 3) a mix of different methodologies e.g. easyOCR, different CNNs without any clear indication of the novelty. Unfortunately, none of these concerns were addressed. The authors were not able to extend the data set. They have tried to data augmentation but it was unsuccessfully (most probably because the dataset is too small). They then implemented class weighting, which is generally not recommended. With respect to mix of different technologies, it was expected that the authors look into latest trends e.g. LLMs, attention mechanisms or a combination of CNNs plus attention in order to come up with a novel and better algorithm for the given problem. Given a very small dataset, all of the claims in the proposed work are superficial. Another important aspect is the applicable area of the proposed work i.e. assisting physicians, which require a very stable and reliable system. Recall is a very important parameter in this context to rule-out false alarms. I still have serious concerns about the novelty, reliability, reproducibility and stability of the proposed approach.

Another issue with the current dataset is class imbalance. Class negative has only 81/543. What strategies have been applied to resolve and mute the impact of class imbalance? Because of train, validate, test split only 323 samples are available for initial training, which will generate statistically insignificant results.

Comments on the Quality of English Language

None

Author Response

We wish to express our sincere gratitude for your valuable comments and suggestions, which have helped to improve the quality of our manuscript. We have significantly revised the manuscript accordingly.

  1. My three major concerns were 1) very small dataset (only 543 images), 2) serious class imbalance (only 81/543 (around 15%) samples are negative), and 3) a mix of different methodologies e.g. easyOCR, different CNNs without any clear indication of the novelty. Unfortunately, none of these concerns were addressed. The authors were not able to extend the data set. They have tried to data augmentation but it was unsuccessfully (most probably because the dataset is too small). They then implemented class weighting, which is generally not recommended. With respect to mix of different technologies, it was expected that the authors look into latest trends e.g. LLMs, attention mechanisms or a combination of CNNs plus attention in order to come up with a novel and better algorithm for the given problem. Given a very small dataset, all of the claims in the proposed work are superficial. Another important aspect is the applicable area of the proposed work i.e. assisting physicians, which require a very stable and reliable system. Recall is a very important parameter in this context to rule-out false alarms. I still have serious concerns about the novelty, reliability, reproducibility and stability of the proposed approach.

Another issue with the current dataset is class imbalance. Class negative has only 81/543. What strategies have been applied to resolve and mute the impact of class imbalance? Because of train, validate, test split only 323 samples are available for initial training, which will generate statistically insignificant results.

[Response]

We appreciate your detailed review and have carefully considered the concerns you raised regarding the small dataset, class imbalance, the mix of different methodologies without clear novelty, and the reproducibility and stability of our approach. We fully understand the importance of these issues and their potential impact on the validity of our findings.

However, we would like to emphasize that the primary objective of our study was not to develop a novel methodology, but rather to demonstrate a proof of concept by integrating existing technologies. We believe that our work contributes to the field by exploring the practical application and potential benefits of combining these methodologies, which could serve as a foundation for future research.

In response to your feedback, we have also ensured that our Discussion section thoroughly addresses both the strengths and weaknesses of our study. By doing so, we aim to provide readers with a clear understanding of the study’s purpose and contributions, while acknowledging the limitations that were identified.

[Revised]

  • Our findings suggest that advanced DL models for classifying SWE-mode images can significantly improve the accuracy and reliability of elasticity classification, thereby enhancing the overall diagnostic process for breast cancer.

While our dataset size is relatively small (543 images) and exhibits class imbalance, it's important to note that this imbalance reflects the real-world distribution of positive and negative cases in clinical settings. Despite these limitations, our findings suggest that advanced DL models for classifying SWE-mode images can potentially improve the accuracy and reliability of elasticity classification, thereby enhancing the overall diagnostic process for breast cancer. However, we acknowledge that a larger and more diverse dataset would be necessary to fully validate these results and ensure their generalizability across different clinical contexts.

  • We believe that the ultimate form of US report generation solutions will likely incorporate elements of both approaches. However, for these solutions to be used in actual clinical practice, many issues still need to be addressed, including improving the accuracy, reliability, and robustness of generated reports.

We acknowledge that our current approach combines different methodologies, including easyOCR and various CNNs. While this may not represent a single novel algorithm, it demonstrates the feasibility of integrating multiple cutting-edge technologies to address a complex clinical challenge. Future research could explore the incorporation of more advanced techniques such as Large Language Models (LLMs) and attention mechanisms to further enhance the system's performance and novelty. We recognize that for clinical applications, especially those assisting physicians, a highly stable and reliable system is crucial. Our current results, while promising, would require further validation and improvement to meet the stringent requirements for clinical use. Future work should focus on enhancing the system's reliability, reproducibility, and stability.

  • our findings were obtained using representative deep learning-based algorithms that demonstrated excellent performance in text extraction and image classification tasks on natural images, rather than algorithms specialized for breast ultrasound scan images.

our study was positioned as a proof of concept, exploring the integration of existing technologies rather than developing novel methodologies. Consequently, we employed well-established deep learning-based algorithms that have proven effective in text extraction and image classification tasks on natural images, rather than algorithms specialized for breast ultrasound scan images.

  • we aim to enhance our system's overall performance by implementing state-of-the-art deep learning architectures, developing specialized models tailored for breast ultrasound image analysis, and fine-tuning these models on larger, more diverse datasets, potentially leading to more accurate and reliable automated breast ultrasound reporting.

we aim to enhance our system's overall performance by developing specialized models tailored for breast ultrasound image analysis, incorporating cutting-edge technologies such as large language models and attention mechanisms in their development.

[Added]

  • While our current results are promising, we recognize the need for further research and development to address the limitations identified in this study and to meet the rigorous standards required for clinical application.

Reviewer 2 Report (Previous Reviewer 2)

Comments and Suggestions for Authors

This paper proposed a semi-automatic method for breast US report generation that blends human oversight with automation. The study implements deep learning algorithms to enhance the report generation process and reduce the burden on radiologists. The generation of the reports has demonstrated high accuracy.

After modifications, the report generation process has become more clear and the future work has been outlined. One detail that might need review is that the chapter number in line 314, 315, 331, and 357 are out of order.

I suggest this manuscript can be published in the journal.

Author Response

Dear Editor and Reviewers;

We wish to express our sincere gratitude for your valuable comments and suggestions, which have helped to improve the quality of our manuscript. We have significantly revised the manuscript accordingly.

  1. This paper proposed a semi-automatic method for breast US report generation that blends human oversight with automation. The study implements deep learning algorithms to enhance the report generation process and reduce the burden on radiologists. The generation of the reports has demonstrated high accuracy.

After modifications, the report generation process has become more clear and the future work has been outlined. One detail that might need review is that the chapter number in line 314, 315, 331, and 357 are out of order.

I suggest this manuscript can be published in the journal.

[Response]

We sincerely thank you for your careful review and constructive feedback.

In addition, as pointed out the reviewer, we have revised the manuscript accordingly.

Round 2

Reviewer 1 Report (Previous Reviewer 1)

Comments and Suggestions for Authors

I still have concerns about the paper, which were highlighted in previous multiple revisions. Given the updated discussion in the manuscript, the paper can be consider for publication.

This manuscript is a resubmission of an earlier submission. The following is a list of the peer review reports and author responses from that submission.

Round 1

Reviewer 1 Report

Comments and Suggestions for Authors

1. Title: "report generation" is way too abstract. Use more specific terms in the title
2. Abstract should be rewritten as it is very fussy and unstructured. Some of the sentences are too long. reported reduced report generation time of 3.8 minutes is with respect to what? accuracy ratio is not a common term in DL, AI. Why other important evaluation parameters such as sensitivity, specificity, recall, precision, f1-score are not reported?

3. Table 1 shows the distribution of the dataset. CNNs, RNNs, LLMs typically require huge amounts of data to rule out overfitting and to obtain satisfactory and reliable output. The objective of this study is very critical as all of the future clinical courses of action will rely on these semi-generated reports so it is crucial to have a very reliable system. With such a small dataset of only 534 images, do you believe that the system will be reliable? My answer will be NO. I will strongly suggest training and validating on a large scale dataset.

4. Another issue with the current dataset is class imbalance. Class negative has only 81/543. What strategies have applied to resolve and mute the impact of class imbalance? Because of train, validate, test split only 323 samples are available for initial training, which will generate statistically insignificant results.

5. Table 1 says 543 samples, whereas line 94 says 534 samples. Correct it.

6. There is no literature review. Some of the papers are cited in the introduction, however, they do not cover the field in depth and breadth. Authors must include a complete section on literature review and cover the field appropriately.

7. There is no standard format for referencing. Year of publication is missing in many references.

8. Only 20 references are there (rest of the three are links to online resources). Refer to point 6 above and improve literature review significantly.

9. How does the ICVF module work? where are the technical details? Section 2.2.1 does not say anything. Reduce the theory and concentrate on the technical details.

10. What is the purpose of algorithm 1? it is simply diverting to procedures; B-Mode, Doppler-mode, SWE-mode. Where are these procedures?

11. Lines 191: why the main objective is discussed again?

12. How does the ARSC module work? There are no technical details, pseudocode or architecture diagrams.

13. Section 2.3.1. is all theory. This is not a contribution of the authors.

14. Section 2.3.2: what are your contributions?

15. The data is not publicly available so the reproducibility of the work is not possible.

16. Train and validate on publicly available large scale data.

17. Overall the presented work is a mix of different methodologies e.g. easyOCR, different CNNs without any clear indication of the novelty. It is unclear why only these methodologies were chosen?

Comments on the Quality of English Language

Moderate editing of English language required

Author Response

We wish to express our sincere gratitude for your valuable comments and suggestions, which have helped to improve the quality of our manuscript. We have significantly revised the manuscript accordingly.

  1. Title: "report generation" is way too abstract. Use more specific terms in the title

[Response]

"Thank you for your insightful comments and suggestion. Following the reviewer's suggestion, we have revised the title to more specifically reflect the content of our study.

[Before]

AI-Assisted Semi-Automatic Report Generation of Multimodal Ultrasound for Breast Cancers

[Revised]

AI-Powered Synthesis of Structured Multimodal Breast Ultrasound Reports Integrating Radiologist Annotations and Deep Learning Analysis

  1. Abstract should be rewritten as it is very fussy and unstructured. Some of the sentences are too long. reported reduced report generation time of 3.8 minutes is with respect to what? accuracy ratio is not a common term in DL, AI. Why other important evaluation parameters such as sensitivity, specificity, recall, precision, f1-score are not reported?

[Response]

Following the reviewer's suggestion, we have revised the abstract accordingly

[Revised, pg. 1]

Breast cancer is the most prevalent cancer among women worldwide. B-mode ultrasound (US) is essential for early detection, offering high sensitivity and specificity without radiation exposure. This study introduces a semi-automatic method to streamline breast US report generation, aiming to reduce the burden on radiologists. Our method synthesizes comprehensive breast US reports by combining the extracted information from radiologists’ annotations during routine screenings with the analysis results from deep learning algorithms on multimodal US images. Key modules in our method include Image Classification using Visual Features (ICVF), Type Classification via Deep Learning (TCDL), and Automatic Report Structuring and Compilation (ARSC). Experiments showed that the proposed method reduced the average report generation time to 3.8 minutes compared to manual processes, even when using relatively low-spec hardware. Generated reports perfectly matched ground truth reports for suspicious masses without a single failure on our evaluation datasets. Additionally, the deep learning-based algorithm, utilizing DenseNet-121 as its core model, achieved an overall accuracy of 0.865, precision of 0.868, recall of 0.847, F1-score of 0.856, and the area under the receiver operating characteristics of 0.92 in classifying tissue stiffness in breast US shear-wave elastography (SWE-mode) images. These improvements not only streamline the report generation process but also allow radiologists to dedicate more time and focus on patient care, ultimately enhancing clinical outcomes and patient satisfaction.

  1. Table 1 shows the distribution of the dataset. CNNs, RNNs, LLMs typically require huge amounts of data to rule out overfitting and to obtain satisfactory and reliable output. The objective of this study is very critical as all of the future clinical courses of action will rely on these semi-generated reports so it is crucial to have a very reliable system. With such a small dataset of only 534 images, do you believe that the system will be reliable? My answer will be NO. I will strongly suggest training and validating on a large scale dataset.

[Response]

Several deep learning-based studies have utilized both B-mode and SWE-mode images to enhance breast cancer diagnostic performance. However, to the best of our knowledge, this is the first study to analyze the feasibility of automatically classifying SWE-mode images based on their corresponding elasticity. Therefore, there is no large-scale public dataset that has been established for this purpose.

However, we strongly agree with the reviewer’s concern that validation using a large-scale dataset is essential for ensuring system reliability. Due to the current lack of such datasets, we acknowledge this as a limitation of our study. Moving forward, we plan to collaborate with medical institutions to build a larger dataset and validate our approach more extensively. We hope the reviewer understands these constraints.

[Revised, pg. 14]

This study has several limitations. First, we used only a single model of equipment for breast US scan image acquisition. This limitation may affect the generalizability of our results to images obtained from different ultrasound machines, potentially leading to decreased performance when applied to diverse clinical settings. Second, we did not evaluate the proposed system or DL models using external datasets, which is crucial for ensuring the system's reliability and generalizability. This lack of external validation limits our ability to confidently claim the system's effectiveness across different patient populations and clinical environments. Third, we did not perform a comprehensive analysis according to various factors such as demographic characteristics and lesion sizes. Fourth, our findings were obtained using representative deep learning-based algorithms that demonstrated excellent performance in text extraction and image classification tasks on natural images, rather than algorithms specialized for breast ultrasound scan images. To address these limitations, future work will focus on collaborating with medical institutions to build a larger and more diverse dataset, allowing for extensive validation of our approach. In addition, we aim to enhance our system's overall performance by implementing state-of-the-art deep learning architectures, developing specialized models tailored for breast ultrasound image analysis, and fine-tuning these models on larger, more diverse datasets, potentially leading to more accurate and reliable automated breast ultrasound reporting.

  1. Another issue with the current dataset is class imbalance. Class negative has only 81/543. What strategies have been applied to resolve and mute the impact of class imbalance? Because of train, validate, test split only 323 samples are available for initial training, which will generate statistically insignificant results.

[Response]

Thank you for your feedback.

We addressed the class imbalance by using data augmentation for the minority class, but this did not significantly improve the results. To ensure reliability, we implemented class weighting during the training process, adjusting the class weights inversely proportional to their frequencies. The information about training is added in section 3.2.6.

[Revised, pg. 10]

Furthermore, our dataset exhibits class imbalance, as shown in Table 1, with the Negative class significantly underrepresented compared to the Equivocal and Positive classes. To address this imbalance, we assigned higher weights to the minority class during training. This ensures that misclassifications of the minority classes are penalized more heavily, encouraging the model to learn more effectively from these classes.

  1. Table 1 says 543 samples, whereas line 94 says 534 samples. Correct it.

[Response]

Thank you for pointing that out.

We have corrected line 94 to accurately reflect the 543 samples as stated in Table 1.

  1. There is no literature review. Some of the papers are cited in the introduction, however, they do not cover the field in depth and breadth. Authors must include a complete section on literature review and cover the field appropriately.

[Response]

Thank you for your valuable feedback. 

Following the reviewer's suggestion, we have included a complete section on literature review for report generation systems for ultrasound scan images.

  1. There is no standard format for referencing. Year of publication is missing in many references.

[Response]

Thank you for your comments.

We have corrected the referencing format to include the year of publication where it was missing.

  1. Only 20 references are there (the rest of the three are links to online resources). Refer to point 6 above and improve literature review significantly.

[Response]

Thank you for your insightful comments and suggestion.

This review is addressed in the point No. 6.

  1. How does the ICVF module work? Where are the technical details? Section 2.2.1 does not say anything. Reduce the theory and concentrate on the technical details.

[Response]

Thank you for your insightful comments and suggestions.

Following the reviewer's suggestion, we have revised Section 2.2.1 to provide the technical details of distinguished features of the ICVF.

[Revised, pg. 5-6]

This module serves as the base as it analyzes US breast scans, systematically identifying their class based on several visual attributes. In this paper, we considered four types of US scans (i.e., B-mode, Doppler elastography, strain elastography and shear-wave elastography). Examples of them are shown in Figure 3. Each type of scan has unique visual attributes that are specific to that type. For instance, in SWE-mode images, distinct characteristics emerged. Notably, the presence of a color bar with accompanying legends denoting units in kilopascals (kPa) and large colored contours were observed. SE-mode images also exhibited notable traits, with absence of kPa unit indicators along the color legend bar, showcasing similarly large contours as observed in SWE-mode images. Conversely, Doppler images were distinguished by relatively smaller colored contour regions in comparison with SE-mode and SWE-mode images. Additionally, a notable feature was the presence of colored regions of interest (ROIs) that did not conform to a rectangular or square shape commonly found in SE-mode and SWE-mode images. These observations collectively aided naked-eye identification and differentiation of visual attributes across different types of elastography images.

To classify US scan images based on their visual characteristics discussed above, an image processing algorithm is introduced, as shown in Figure 4. The algorithm first checks if the image is grayscale; if so, it categorizes the image as B-mode. For non-grayscale images, it assesses color pixel density and detects color contours to determine the mode. If the color pixel density is low and the contours are irregular, the image is classified as Doppler-mode. Conversely, if the text "kPa" is detected within the image around a colored legend bar, it is classified as SWE-mode. If neither condition is met, the algorithm defaults to SE-mode. This classification process utilizes a combination of color analysis, text detection, and contour examination to accurately determine the ultrasound imaging mode. Lastly, this module outputs an image annotated with its associated class, which is then passed as input to the Type Classification via Deep Learning (TCDL) module and Automatic Report Structuring and Compilation (ARSC) module for further parallel processing.

  1. What is the purpose of algorithm 1? it is simply diverting to procedures; B-Mode, Doppler-mode, SWE-mode. Where are these procedures?

[Response]

Algorithm 1, the core component of the ICVR module, classifies input ultrasound scan images into four distinct types based on their visual attributes: B-mode, Doppler elastography, Strain elastography (SE-mode), and Shearwave elastography (SWE-mode). This classification is crucial as each type exhibits unique visual characteristics, necessitating specialized analysis approaches.

Our method leverages this classification to apply dedicated deep learning models for each image type, rather than using a single combined model. This targeted approach is designed to enhance the accuracy and depth of analysis. By invoking specific models tailored to the visual nuances of each ultrasound scan type, we expect to achieve more precise and comprehensive analytical results compared to a one-size-fits-all model approach.

  1. Lines 191: why the main objective is discussed again?

[Response]

Thank you for your valuable feedback.  

We have revised the sentences accordingly.

[Revised, pg. 6]

Then the ARSC module formulates descriptive sentences for each group, encapsulating key finding. The template sentence was defined for the section of findings to ensure systematic and consistent reporting:

  1. How does the ARSC module work? There are no technical details, pseudocode or architecture diagrams.

[Response]

Thank you for your insightful comments and suggestions.

Following the reviewer's suggestion, we have revised Section 2.2.3 to provide the technical details of distinguished features of the ARSC. In addition, we also have included the architecture diagram of the module.

[Revised, pg. 7]

The ARSC module is an integral part of automatic report generation. It begins with preprocessing, where it receives inputs from the TDCL module in the form of images, class, and associated scan types. As shown in Figure 6, the ARSC module consists of two main steps: Text Extraction and Report Compilation. In the Text Extraction, images from the ICVF module undergo pre-processing to isolate relevant text, located at the bottom third of DICOM scans. Hence, partial images containing only relevant textual parts were extracted from original US scans. The Optical Character Recognition (OCR) algorithm then extracted textual information from these partial images. Thus, sorting and cleaning were needed to ensure accuracy. The location and size information were extracted using bounding box details, which might contain inaccuracies. A context-based correction approach was applied to correct these inaccuracies, such as replacing misclassified characters like 'S' with '5', 'I' with '1', 'Z' with '2', and 'b' with '6' within size-related information. We developed a character confusion matrix to identify and correct common OCR misrecognitions, assigning context-based weights or ranks to each confusion pair based on their frequency. This matrix was developed during the testing and validation phase, comparing OCR outputs with ground truth data provided by radiologists. For instance, the correction of 'S' to '5' is given higher weight due to its frequent occurrence, especially when 'cm' follows, as 'cm' in our dataset is consistently followed by a number. These corrections, weighted by their frequency, have significantly improved the OCR performance, reducing the Character Error Rate (CER) from 25% to 0%.  The cleaning process refined extracted data, ensuring consistency and precision. Only decimal numbers were retained for size information. The second step, Report Compilation, involves grouping scans based on location data using inputs from TDCL Module. Afterwards, refined text and scan type information were used to group scans based on location data.

[Added, pg. 8]

  1. Section 2.3.1. is all theory. This is not a contribution of the authors.

[Response]

Thank you for your valuable feedback.

Our method synthesizes comprehensive breast US reports by combining information extracted from radiologists' annotations during routine screenings with analysis results from deep learning algorithms on multimodal US images. In this context, text extraction and character recognition are critical technologies in our work. Section 2.3.1 serves to provide essential background for reader comprehension of these technologies.

As the reviewer correctly pointed out, our current implementation utilizes existing state-of-the-art solutions for text extraction and character recognition. While this may not represent a significant contribution in that specific area, our study presents an important intermediate approach in the evolution from fully manual report generation to fully automated analysis and report generation based on ultrasound images.

Our primary contribution lies in presenting a methodology that leverages these algorithms to semi-automatically synthesize structured reports, rather than in improving the performance of optical character recognition (OCR) algorithms themselves. This approach bridges the gap between manual and fully automated processes, offering a practical solution for clinical settings.

However, we acknowledge that the performance of OCR algorithms, which excel on natural images, may not necessarily translate to ultrasound scan images acquired from various machines. This potential limitation has informed one of our future research directions: developing robust OCR algorithms specifically for ultrasound images from different machines. We have included this aspect in our future work section to address this important consideration.

[Revised, pg. 14]

Fourth, our findings were obtained using representative deep learning-based algorithms that demonstrated excellent performance in text extraction and image classification tasks on natural images, rather than algorithms specialized for breast ultrasound scan images. To address these limitations, future work will focus on collaborating with medical institutions to build a larger and more diverse dataset, allowing for extensive validation of our approach. In addition, we aim to enhance our system's overall performance by implementing state-of-the-art deep learning architectures, developing specialized models tailored for breast ultrasound image analysis, and fine-tuning these models on larger, more diverse datasets, potentially leading to more accurate and reliable automated breast ultrasound reporting.

  1. Section 2.3.2: what are your contributions?

[Response]

Thank you for your valuable feedback.

This is the first study to analyze the feasibility of automatically classifying breast shearwave elastography images based on their corresponding elasticity. While this study did not present a self-developed deep learning algorithm for classification, we believe that it makes a significant contribution by demonstrating the potential for automatic classification of breast shearwave elastography image types. This capability is particularly valuable because it can be applied various application including automated report generation, breast cancer diagnosis, and etc.

From this perspective, Section 2.3.2 was written to introduce the main operational processes of representative deep learning models used for image classification in this study, aiming to enhance readers' comprehension of the subject matter.

However, by developing deep learning models specialized for each type of breast ultrasound elastography image, we believe that we can achieve improved performance compared to the models used in the current study. Therefore, the development of more sophisticated deep learning models has been included as a part of future research.

[Revised, pg. 14]

Fourth, our findings were obtained using representative deep learning-based algorithms that demonstrated excellent performance in text extraction and image classification tasks on natural images, rather than algorithms specialized for breast ultrasound scan images. To address these limitations, future work will focus on collaborating with medical institutions to build a larger and more diverse dataset, allowing for extensive validation of our approach. In addition, we aim to enhance our system's overall performance by implementing state-of-the-art deep learning architectures, developing specialized models tailored for breast ultrasound image analysis, and fine-tuning these models on larger, more diverse datasets, potentially leading to more accurate and reliable automated breast ultrasound reporting.

  1. The data is not publicly available, so the reproducibility of the work is not possible.

[Response]

Thank you for your insightful comments and suggestions.

While the data is property of the hospital and requires consent for sharing, we are willing to provide data and code to facilitate reproducibility upon request. Details regarding data availability are provided on page 15.

  1. Train and validate on publicly available large scale data.

[Response]

Thank you for your insightful comments and suggestion. This response is addressed in the point No. 3

  1. Overall the presented work is a mix of different methodologies e.g. easyOCR, different CNNs without any clear indication of the novelty. It is unclear why only these methodologies were chosen?

[Response]

Thank you for your valuable feedback.

As an intermediate step toward the evolution of fully automated report generation, this study proposed a semi-automated approach that could blend human oversight with automation. This approach leverages existing annotations to generate breast reports, thereby reducing radiologists' workload while maintaining high accuracy. Additionally, this study implements deep learning algorithms to classify tissue stiffness in breast US SWE-mode images, enhancing the report generation process.

The selection of methodologies such as EasyOCR, MMOCR, Pytesseract Library, and East Detector was based on their capabilities and suitability for our specific application. Among these, EasyOCR demonstrated superior performance in accurately recognizing text and characters in our experiments, hence it was chosen as the primary OCR tool in our study. Additionally, for classification tasks, we exclusively utilized state-of-the-art and benchmark CNN models (ResNet-50, DenseNet-121, and EfficientNet-B6), comparing their performance. Our findings highlight DenseNet-121 as the top performer in terms of classification accuracy and other relevant metrics.

Our primary contribution lies in presenting a methodology that leverages these algorithms to semi-automatically synthesize structured reports, rather than in improving the performance of individual deep learning-based algorithms themselves. This approach bridges the gap between manual and fully automated processes, offering a practical solution for clinical settings.

However, as part of our future research, we plan to improve the deep learning models used at each stage of the report generation pipeline, and we have included this in the Discussion section.

Reviewer 2 Report

Comments and Suggestions for Authors

This paper proposed a semi-automatic method for breast US report generation that blends human oversight with automation. The study implements deep learning algorithms to enhance the report generation process and reduce the burden on radiologists. The generation of the reports has demonstrated high accuracy. However, the current version still contains quite a few concerns.

1. In line 52, the format of the citation is different from the citation on line 33. The format should be regularized.

2. In Section 2.2.1, it is mentioned that the ICVF module can only classify four types of US scans (B-mode, Doppler elastography, Strain elastography and Shearwave elastography). If an image of another type is input into the ICVF, will it still work? Sometimes the image sample size is large and other types of images may be accidentally mixed in. What will the ICVF module output?

3. From line 141 to line 156, the article describes the characteristics of the four types of images. This method of classification relies on the visual features of the images. However, if the ultrasound image is from another agency or device, the visual features may be different, which will lead to the performance degradation of the classification algorithms. This classification method is too limited to be used flexibly in other application scenarios.

4. In line 182, it is mentioned that “conditions applied to correct ‘S’ with ‘5’, ‘I’ with ‘1’, ‘Z’ with ‘2’, and ‘b’ with ‘6’”. Specific conditions of distinguishing these characters are not described in the following section, and they should be explained clearly. The performance of OCR should also be tested and evaluated.

Author Response

We wish to express our sincere gratitude for your valuable comments and suggestions, which have helped to improve the quality of our manuscript. We have significantly revised the manuscript accordingly.

This paper proposed a semi-automatic method for breast US report generation that blends human oversight with automation. The study implements deep learning algorithms to enhance the report generation process and reduce the burden on radiologists. The generation of the reports has demonstrated high accuracy. However, the current version still contains quite a few concerns.

  1. In line 52, the format of the citation is different from the citation on line 33. The format should be regularized.

[Response]

Thank you for your insightful comments and suggestion.

The format has been regularized.

  1. In Section 2.2.1, it is mentioned that the ICVF module can only classify four types of US scans (B-mode, Doppler elastography, Strain elastography and Shearwave elastography). If an image of another type is input into the ICVF, will it still work? Sometimes the image sample size is large and other types of images may be accidentally mixed in. What will the ICVF module output?

[Response]

Thank you for your comments.

Currently, our testing is limited to a specific dataset from a hospital, ensuring the ICVF module processes only defined types of ultrasound scans. If other types of images are inadvertently included, the module would classify them as "Others." Future scalability could involve adjustments like if-else conditions to handle unknown image types effectively but this is not the part of current work and can be addressed in future work.

  1. From line 141 to line 156, the article describes the characteristics of the four types of images. This method of classification relies on the visual features of the images. However, if the ultrasound image is from another agency or device, the visual features may be different, which will lead to the performance degradation of the classification algorithms. This classification method is too limited to be used flexibly in other application scenarios.

[Response]

Thank you for your insights.

We acknowledge that relying on data from a single hospital and device limits the generalizability of our classification method to ultrasound scan images from other agencies or devices. These related issues have been noted as limitations of our study. To address these limitation, future work will focus on expanding the dataset, integrating multi-center data, and further improving the system's robustness and accuracy to ensure clinical applicability.

[Revised, pg. 14]

This study has several limitations. First, we used only a single model of equipment for breast US scan image acquisition. This limitation may affect the generalizability of our results to images obtained from different ultrasound machines, potentially leading to decreased performance when applied to diverse clinical settings. Second, we did not evaluate the proposed system or DL models using external datasets, which is crucial for ensuring the system's reliability and generalizability. This lack of external validation limits our ability to confidently claim the system's effectiveness across different patient populations and clinical environments. Third, we did not perform a comprehensive analysis according to various factors such as demographic characteristics and lesion sizes. Fourth, our findings were obtained using representative deep learning-based algorithms that demonstrated excellent performance in text extraction and image classification tasks on natural images, rather than algorithms specialized for breast ultrasound scan images. To address these limitations, future work will focus on collaborating with medical institutions to build a larger and more diverse dataset, allowing for extensive validation of our approach. In addition, we aim to enhance our system's overall performance by implementing state-of-the-art deep learning architectures, developing specialized models tailored for breast ultrasound image analysis, and fine-tuning these models on larger, more diverse datasets, potentially leading to more accurate and reliable automated breast ultrasound reporting.

  1. In line 182, it is mentioned that “conditions applied to correct ‘S’ with ‘5’, ‘I’ with ‘1’, ‘Z’ with ‘2’, and ‘b’ with ‘6’”. Specific conditions of distinguishing these characters are not described in the following section, and they should be explained clearly. The performance of OCR should also be tested and evaluated.

[Response]

Thank you for your insightful feedback.

Following the reviewer's suggestion, we have revised the relevant section accordingly

[Revised, pg. 7]

We developed a character confusion matrix to identify and correct common OCR misrecognitions, assigning context-based weights or ranks to each confusion pair based on their frequency. This matrix was developed during the testing and validation phase, comparing OCR outputs with ground truth data provided by radiologists. For instance, the correction of 'S' to '5' is given higher weight due to its frequent occurrence, especially when 'cm' follows, as 'cm' in our dataset is consistently followed by a number. These corrections, weighted by their frequency, have significantly improved the OCR performance, reducing the Character Error Rate (CER) from 25% to 0%.
